Proceedings of Machine Learning Research 1–12, 2021                         Full Paper – MIDL 2021

# Stroke Lesion Outcome Prediction Based on 4D CT Perfusion Data Using Temporal Convolutional Networks

**Kimberly Amador**[1]                                  KIMBERLYALEJANDRA.AM@UCALGARY.CA

**Matthias Wilms**[1,2]                              MATTHIAS.WILMS@UCALGARY.CA

**Anthony Winder**[1]                              ANTHONY.WINDER@UCALGARY.CA

**Jens Fiehler**[3]                                    FIEHLER@UKE.DE

**Nils D. Forkert**[1,2,4]                             NILS.FORKERT@UCALGARY.CA

[1] *Department of Radiology and Hotchkiss Brain Insitute, University of Calgary, Calgary, Canada*

[2] *Alberta Children's Hospital Research Institute, University of Calgary, Calgary, Canada*

[3] *Department of Diagnostic and Interventional Neuroradiology, University Medical Center Hamburg-Eppendorf, Hamburg, Germany*

[4] *Department of Clinical Neuroscience, University of Calgary, Calgary, Canada*

## Abstract

Acute ischemic stroke is caused by a blockage in the cerebral arteries, resulting in long-term disability and sometimes death. To determine the optimal treatment strategy, a patient-specific assessment is often based on advanced neuroimaging data, such as spatio-temporal (4D) CT Perfusion (CTP) imaging. To date, perfusion maps are typically calculated from 4D CTP data and then thresholded to localize and quantify the stroke lesion core and tissue-at-risk. A few studies have recently developed deep learning methods to predict stroke lesion outcomes from perfusion maps. The basic idea of these is to train a model, using perfusion maps acquired at baseline and their corresponding follow-up images acquired several days after treatment, to automatically estimate the final lesion location and volume in new patients. Nevertheless, model training based on the original 4D CTP scans might be desirable, as they could contain more valuable information not directly represented in perfusion maps. Therefore, we aimed to develop and evaluate a temporal convolutional neural network (TCN) to predict stroke lesion outcomes directly from 4D CTP datasets acquired at admission, without computing any perfusion maps. Using a total of 176 CTP scans, we investigated the impact of the time window size by training the proposed TCN on various numbers of CTP frames: 8, 16, and 32 time points. For comparison purposes, we also trained a convolutional neural network based on perfusion maps. The results show that the model trained on 32 time points yielded significantly higher Dice values ($0.33\pm0.21$) than the models trained on 8 time points ($0.25\pm0.20$; $P<0.05$), 16 time points ($0.28\pm0.21$; $P<0.001$), and perfusion maps ($0.23\pm0.18$; $P<0.05$). These experiments demonstrate that the proposed model effectively extracts spatio-temporal data from CTP scans to predict stroke lesion outcomes, which leads to better results than using perfusion maps.

**Keywords:** stroke, outcome prediction, CT perfusion, deep learning, temporal convolutional networks

## 1. Introduction

Acute ischemic stroke (AIS), a disorder caused by a blockage in the cerebral blood flow, is a leading cause of death and the largest source of acquired disability worldwide (Feigin et al., 2019). "Time is brain" when diagnosing and treating AIS: the longer therapy is delayed, and

thus blood flow is not restored, the more tissue may become irreversibly damaged (Saver, 2006). Therefore, access to the most efficient treatment options and the ability to make rapid treatment decisions are two critical factors in determining a patient's outcome.

To determine the optimal treatment strategy, a patient-specific assessment is typically performed based on clinical information and advanced neuroimaging data (Mokli et al., 2019). Specifically, spatio-temporal (4D) Computed Tomography Perfusion (CTP) has been successfully incorporated into AIS protocols to assess the blood flow situation in the brain as it is faster, less expensive, and more widely available compared to other medical imaging techniques (Yu et al., 2016). In 4D CTP imaging, a spatio-temporal series of 3D images are acquired (for about 60 s) after the intravenous injection of a contrast agent to identify areas with abnormal perfusion, including irreversibly damaged tissue (infarct core) and tissue-at-risk (penumbra) (Demeestere et al., 2020). Nevertheless, 4D CTP datasets are difficult to analyze due to the vast amount of data and low signal-to-noise ratio (SNR) of the images, in addition to the complex appearance of stroke lesions.

In current clinical practice, perfusion parameter maps (i.e., cerebral blood flow (CBF), cerebral blood volume (CBV), mean transit time (MTT), and Tmax) are calculated from 4D CTP scans and then thresholded to localize and quantify the infarct core and penumbra (Laughlin et al., 2019). The calculation of these relies on the appropriate selection of the so-called arterial input function (AIF), which is needed to correct different contrast agent injection protocols and differences due to patient-individual cardiac output function (Winder et al., 2020). Any under- or overestimation of the AIF can lead to incorrect perfusion parameter values. Furthermore, using simple thresholds to guide treatment decision-making oversimplifies the highly heterogeneous complexity of the AIS (Nielsen et al., 2018).

With the success of deep learning in medical imaging (Lo Vercio et al., 2020), new methods have been proposed to predict the stroke lesion outcome from perfusion maps. The basic idea is to train a model, using perfusion maps acquired at baseline prior to any treatment and their corresponding follow-up images acquired several days after treatment, to predict the final lesion location and volume in new patients. Any observed lesion growth is a result of several factors such as type of treatment, success of the treatment, and the time from baseline imaging to treatment. Although these methods have led to improved results over simple thresholding (Clèrigues et al., 2019), perfusion maps remain noise-sensitive and AIF dependent, limiting further improvements in the prediction accuracy of stroke lesions (Wang et al., 2020). Moreover, these might not display all relevant information available in 4D CTP datasets. Thus, improved tissue outcome prediction results could be achieved if the training was done directly on the 4D CTP scans instead of the perfusion maps.

The aim of this work was to train a novel spatio-temporal deep learning model to predict stroke lesion outcomes directly from 4D CTP datasets. Particularly, a temporal convolutional network (TCN) was developed and evaluated for this purpose, as it has shown promising results in other medical image analysis problems (Krebs et al., 2020) but has never been applied for image analysis in AIS. TCNs are a variation of convolutional neural networks (CNN) that aim to build a hierarchy in a tree-like fashion for combining temporal information from neighboring images. They are less computationally expensive and less memory intensive than conventional CNNs (Bai et al., 2018) and, therefore, they are ideal for training on large time-series data, such as 4D CTP scans.

## 2. Related Work

While automatic segmentation of stroke lesions has been extensively explored (Clèrigues et al., 2019; Tuladhar et al., 2020; Rekik et al., 2012), fewer approaches have been proposed for predicting acute ischemic stroke lesion outcomes from acute imaging and clinical data. Most of them are based on Magnetic Resonance (MR) imaging (Pinto et al., 2018a; Winder et al., 2019), primarily using two sequences: MR perfusion, which is similar to CTP, and diffusion-weighted imaging (DWI), which is considered the gold standard for identifying the anatomic extent of the stroke lesion core due to its good soft tissue contrast (Lansberg et al., 2000). Nonetheless, CTP imaging is faster and more widely available than MR in the clinical environment (Yu et al., 2016), motivating the development of a reliable decision support model based on this type of data.

Due to the low contrast and low SNR of CTP scans, it has been challenging to directly use these images for ischemic stroke lesion prediction tasks. Kasasbeh et al. (2019) were the first to use artificial neural networks (ANN) to predict the stroke lesion core from CTP-derived perfusion maps. Moreover, only a handful of studies have tried to exploit the spatio-temporal nature of raw perfusion data for tissue outcome prediction. For example, Giacalone et al. (2018) demonstrated the added value of using raw MR perfusion data by encoding the spatio-temporal signature of each voxel using a standard support vector machine classifier. Pinto et al. (2018b) proposed a deep learning method for the ISLES 2017 challenge that used raw MR perfusion data, along with DWI images and perfusion maps, to improve the prediction of the tissue outcome. The ISLES 2017 challenge (Winzeck et al., 2018) aimed to compare methods for predicting stroke lesion outcomes based on acute MR imaging data. The winning model achieved a mean Dice score of 0.32, which merely emphasizes the inherent difficulty of the stroke outcome prediction problem. More recently, Robben et al. (2020) demonstrated that working with raw CTP data can lead to better results compared to using perfusion maps. Briefly described, they trained a CNN on patient-specific treatment parameters and native CTP measurements, which consisted of time-attenuation curves of a voxel, its neighboring voxels, and the AIF. The prediction accuracy of their model is highly dependent on the AIF selection, which was performed manually in all cases. Nevertheless, selecting a good AIF is a very challenging problem. Even though some of these preliminary works highlight the potential of spatio-temporal data in stroke outcome prediction, an appropriate method to handle the temporal information, without the need for selecting and including an AIF, is still needed.

## 3. Methods

The main goal of this work was to design a deep neural network that extracts spatio-temporal features from CTP datasets and predicts the tissue outcome as a spatial output (lesion segmentation). For this study, we decided to process each 2D slice of the 4D CTP datasets independently. The rationale behind this is that training a deep learning model on a simpler representation, such as 2D images, is less memory intensive and less computationally expensive compared to using 3D volumetric data (Singh et al., 2020). In addition, current state-of-the-art approaches (Nielsen et al., 2018; Pinto et al., 2018a) mainly use 2D networks to predict the stroke lesion outcome, further encouraging the use of 2D slices for the present study.

### 3.1. Model architecture

The proposed framework for AIS outcome prediction can be divided into three modules (Figure 1a): encoder, temporal convolutional network (TCN), and decoder. Let us assume we are given an input sequence with $T$ time points. First, a series of encoders $E$ with the same structure $(E_0, E_1, ..., E_T)$ are built in parallel to independently map each image $I$ of the sequence $(I_0, I_1, ..., I_T)$ to a latent space. These latent vectors are concatenated and then passed into the TCN, preserving its temporal location. Consisting of multiple 1D layers with increasing dilation, the TCN merges the information across the different time points available. Finally, the decoder takes the TCN output and reconstructs the image, predicting the probability of infarction at each voxel. A binary lesion mask is later generated from the probability maps using an optimized probability threshold $k$. Details related to the network structure are described in Appendix A.

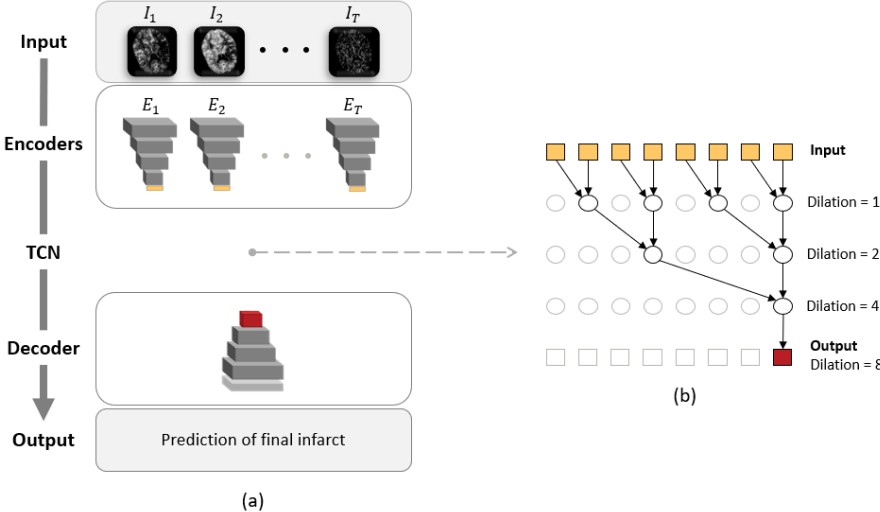

Figure 1: (a) Schematic representation of the proposed model for stroke lesion outcome prediction. (b) The TCN combines temporal information from neighboring images through a series of dilated convolutions.

### 3.1.1. TEMPORAL CONVOLUTIONAL NETWORK

Temporal convolutional networks have shown to be capable of outperforming conventional CNNs and recurrent neural networks in various sequence modelling tasks (Bai et al., 2018). This type of architecture provides a simple, unified approach to capture spatio-temporal information hierarchically by using causal convolutions, dilated convolutions, and residual connections. The causal convolutions help avoid any information leakage from future to past, meaning that the output at time point $T$ is influenced only by elements from that time point and earlier in the previous layer. The dilated convolutions help to expand the receptive field of the network with less computational effort. As exemplified in Figure 1b,

the number of dilations increases exponentially with the depth of the network. Adding residual connections to the network helps stabilize deeper and larger TCNs, as they allow layers to learn modifications to the identity mapping rather than the entire transformation. Another distinguishing characteristic of TCNs is that they can take a sequence of any length and return an output sequence of the same length. Nevertheless, we were only interested in predicting a single binary lesion mask and, therefore, the TCN module was configured to return only the last element in the output sequence (Figure 1b).

### 3.1.2. Loss function

Within the dataset, the stroke lesion often occupies only a very small region of the scan, which means that there are more *background* pixels than *lesion* pixels in the image. This imbalanced nature may lead to sub-optimal results due to local minimas, causing the over-prediction of the background class (Zou et al., 2004). Therefore, we implemented a loss function based on the Dice Similarity Coefficient (DSC), which measures the proportion of overlap between two images. The DSC value ranges from 0 to 1, where 1 indicates complete spatial overlap. The DSC between two binary volumes can be written as

$$DSC = \frac{2|X \cap Y|}{|X| + |Y|} \tag{1}$$

where X and Y represent the predicted and ground truth binary lesion volumes, respectively.

## 4. Experiments and Results

### 4.1. Patient data

A total of 178 CTP scans of acute stroke patients from multiple centers were used to develop and evaluate the proposed model for stroke outcome prediction, each of which has a registered follow-up scan and corresponding lesion segmentation available. These scans were pooled together from the prospective cohort studies PRoveIT (Menon et al., 2015) and ERASER (Fiehler et al., 2019), as well as from data acquired at the University Medical Center Hamburg-Eppendorf, Germany, from June 2015 to May 2019. Follow-up scans (based on either non-contrast CT or MR) were acquired between 30 hours and seven days after stroke symptom onset. The follow-up lesions of each dataset were manually segmented semi-automatically by different experienced medical experts using AnToNIa (Forkert et al., 2014) and ITK-SNAP (Yushkevich et al., 2006) software tools. All scans have a spatial resolution of 512×512×16 voxels and an inter-slice spacing of 5 mm. The dataset was randomly divided (based on the follow-up lesion volume) into a training set (133 patients), validation set (28 patients), and test set (17 patients) using a 75/15/10 split.

### 4.2. Preprocessing

The preprocessing, which aims to reduce the noise and variation of CTP scans, was applied to all scans in five steps: motion correction, baseline correction, temporal smoothing and interpolation, time window selection, and normalization. The first three steps were performed using AnToNIa, a software tool specifically developed for AIS patient evaluation.

Briefly explained, the correction of patient motion was achieved by applying 3D rigid registrations, where each time point was registered to the first one in the sequence. For the baseline correction, an average CTP scan of the first three time points was calculated and then subtracted from all following time points. Since the temporal resolution varies between patients, a temporal interpolation to 1 second per data frame was applied using a b-spline approximation, which also led to a temporal smoothing. Subsequently, temporal sequences of 8, 16, and 32 seconds were extracted from the original CTP scans and then normalized to a zero-mean, unit-variance space across all selected time points. This time window selection was based on the mean intensity-time curve of each scan. First, the average intensity was calculated for each time point. Different sequences were then extracted using the time point with the highest average intensity and its preceding one as mid-points.

Moreover, to avoid processing irrelevant pixels (i.e., background) and further decrease the computation complexity, all input images were cropped to a fixed size of 384×384 pixels. The preprocessed CTP scans were used as inputs, and the lesion segmentations were considered the ground truth for the model.

### 4.3. Implementation details

In the present work, two experiments were performed to 1) explore the effect of the number of frames used on the model performance and 2) investigate the added predictive potential of 4D CTP data compared to perfusion maps. For the first experiment, three models were trained using various sequence lengths: 8 time points (8T), 16 time points (16T), and 32 time points (32T). While the model configuration remains the same, the number of encoders (section 3.1) does change, as it is directly proportional to the number of time points included in the sequence. For the second experiment, a modified version of the proposed model was trained using the perfusion maps (CBF, CBV, MTT, Tmax) and CTP baseline average as input. These perfusion maps were obtained by deconvolving the time-concentration curve in each voxel with the AIF using AnToNIa. The modification to the model consisted of removing the TCN module, as no temporal sequences are present in the input features, resulting in an encoder-decoder (E-D) architecture. All models were trained for 150 epochs using Adam optimizer, a batch size of 1, and a step-based learning rate decay. The above experiments were implemented in Python, using Keras (Chollet et al., 2015) with TensorFlow backend on an NVIDIA Tesla V100 GPU (16 GB memory).

The predictive performance of the models was evaluated using the DSC and absolute volume error between the predicted and ground truth lesions. Both measures were calculated using thresholded lesion probability maps. To determine the optimal threshold $k$, we evaluated the segmentation results at intervals of 0.01 over the whole range [0, 1]. We found that a threshold of $k$=0.48 produced the highest mean DSC for the majority of the trained models. Since the proposed method processes the CTP scans as 2D slices + time, the predicted lesions were post-processed to a 3D space for the quantitative evaluation.

### 4.4. Results and discussion

All evaluation results in Table 1 are reported as mean ± standard deviation, and a representative set of predictions is shown in Figure 2. The highest DSC was achieved by the 32T model (0.33±0.21), followed by the 16T model (0.28±0.21), 8T model (0.25±0.20), and

E-D model (0.23±0.18). Using a paired sample $t$-test, a significant difference was found between the 32T model and 16T model (P<0.001), 8T model (P<0.05), and E-D model (P<0.05). Despite the significant differences in Dice values, no significant difference was found between the absolute lesion volume errors.

These experiments demonstrate that increasing the time window size is beneficial, which can be expected since including more time points provides the model with more information. Although not directly comparable due to different experimental setups and databases, the Dice value achieved by the 32T model (DSC=0.33) is in the range of previously published results (Winzeck et al., 2018). Therefore, we can assume that the proposed architecture can make better use of the spatio-temporal information available in 4D CTP datasets to improve the stroke outcome predictions, especially when compared to simple perfusion maps. It is also worth noting that no explicit or implicit deconvolution of the tissue signal with the AIF was required to predict the final lesion outcome. This characteristic gives our model an important advantage given that deconvolution is a mathematically ill-posed problem.

| Model | DSC | | Absolute Volume Error |
|---|---|---|---|
| E-D | $0.2387 \pm 0.18$ | (*) | $49.50 \pm 29$ ml |
| 8T | $0.2564 \pm 0.20$ | (*) | $32.25 \pm 27$ ml |
| 16T | $0.2836 \pm 0.21$ | (**) | $64.48 \pm 54$ ml |
| 32T | $\mathbf{0.3361 \pm 0.21}$ | | $52.04 \pm 46$ ml |

Table 1: Error metrics comparing the predicted and ground truth lesions. A paired significance test was performed between the proposed method and its variants, with (*) indicating P<0.05 and (**) indicating P<0.001.

| Model | Computation Time | Trainable Parameters |
|---|---|---|
| E-D | 84 s | 10E+06 |
| 8T | 32 s | 54E+06 |
| 16T | 57 s | 104E+06 |
| 32T | 107 s | 205E+06 |

Table 2: Comparison of the model size and computation time at inference for the different implementations.

We also compared the model size and computation time at inference for the different implementations (Table 2). We found that when more time points are added, the longer it takes to predict the lesion outcomes for the whole test set and more trainable parameters are required. Since predicting the stroke lesion outcome in a timely fashion is essential, a downside of the proposed method with 32 time points is the increased memory requirements and computational time at inference.

Our future projects will focus on improving the model performance while generating treatment-specific predictions. Since processing 2D slices independently does not exploit the intrinsic geometric properties of 3D volumetric data, we will explore the impact of spatial

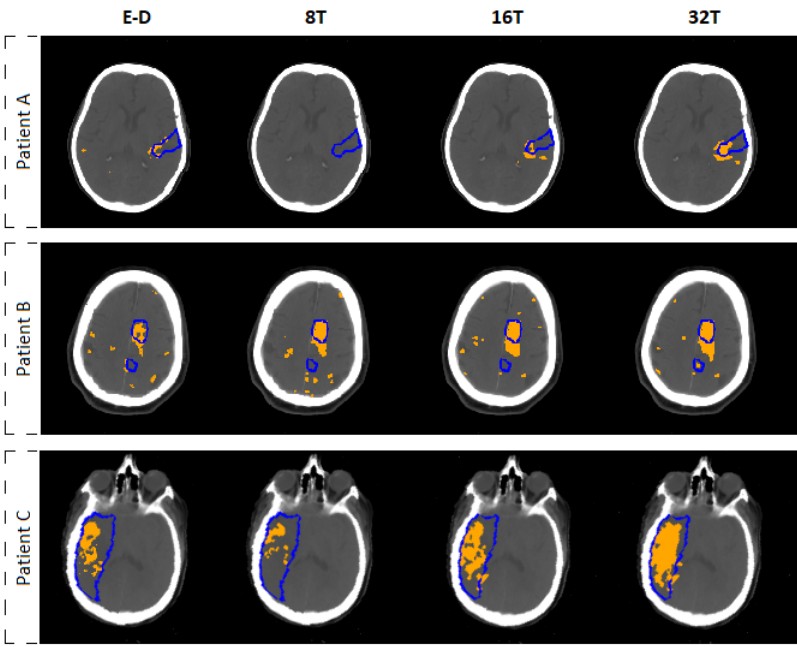

Figure 2: Predictions of the proposed model on a representative sample of subjects from the test set. The thresholded lesion outcome predictions are overlayed in orange on the follow-up scan, and the ground truth is outlined in blue.

context in stroke outcome prediction by training the proposed model on 3D sequences. Additional post-processing steps, such as those used in Tuladhar et al. (2020), will also be included in our next experiments to remove small lesion components presumably caused by noise artifacts.

## 5. Conclusion

In this paper, we proposed a novel deep learning framework for predicting acute ischemic stroke outcomes from 4D CTP datasets. We designed a modular architecture composed of an encoder, decoder, and temporal convolutional network, the latter being key for merging information across different time points. The results show that using longer sequences improves the final infarct prediction, implying that the proposed model effectively learns from spatio-temporal data and leads to better results compared to using perfusion maps.

## Acknowledgments

This work is supported by the Alberta Innovates Graduate Student Scholarships for Data-enabled Innovation program (GSS-DEI), the Canada Research Chairs Program, and the River Fund at Calgary Foundation.

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

## Appendix A. Implementation Details

| Convolutional layers | Number | 10 |
|---|---|---|
| | Kernel size | 3 |
| | Filters | 2·32, 2·64, 2·128 2·256, 2·512 |
| | Stride | 1 |
| Downsampling layers | Number | 5 |
| | Type | Convolution |
| | Kernel size | 2 |
| | Stride | 2 |

Table 3: Implementation details from the 1st module (encoder).                    .

| | | 8T | 16T | 32T |
|---|---|---|---|---|
| Convolutional layers | Dilations | [1, 2, 4] | [1, 2, 4, 8] | [1, 2, 4, 8, 16] |
| | Kernel size | | 2 | |
| | Filters | | 64 | |
| | Stride | | 1 | |

Table 4: Implementation details from the 2nd module (TCN).                    .

| Convolutional layers | Number | 11 |
|---|---|---|
| | Kernel size | 3 |
| | Filters | 2·512, 2·256, 2·128 2·64, 2·32, 2 |
| | Stride | 1 |
| Upsampling layers* | Number | 5 |
| | Type | Convolution |
| | Kernel size | 2 |
| | Stride | 2 |

Table 5: Implementation details from the 3rd module (decoder).
*Skip connections were added at every upsampling step.

