# OpenReview forum: "Stroke Lesion Outcome Prediction Based on 4D CT Perfusion Data Using Temporal Convolutional Networks"
_MIDL.io/2021/Conference — MIDL 2021_

### Official Review · ~Alberto_Gomez1 · 2021-03-05

**Confidence:** 4
**Preliminary Rating:** 3
**Recommendation:** Oral
**Final Rating:** 4

**Summary:**

The authors propose a framework to predict lesion outcome from 4D CT perfusion data with no need to compute perfusion maps. The concept is novel and sound, and although the methodology essentially re-uses architectures previously published, the use of these architectures to this problem is original and very promising.

**Strengths:**

* The clinical challenge is relevant, and described very solidly
* Estimating outcomes directly from CTP without the need of computing perfusion maps is novel and extremely useful if and when it works.
* The paper is very well written.

**Weaknesses:**

* The reported DSC values seem very low (the highest is 0.33). This to me looks like a model that fails. However I acknowledge that for the specific application here this may not necfessarily be the case, but then authors need to explain what a DSC of 0.3 shows. For example, how is this DSC value related to the quality of the predicted outcome? Further, since they are providing example images in Fig. 2, what is the DSC for those images?

* The results fail to convince me that the method actually works. The results show that using longer sequences improve predictions, however there is no justification that even the best performing results are good enough .

**Deanonymize Review:**

yes

**Detailed Comments:**

"to avoid processing irrelevant pixels (i.e., background) and further decrease the computation complexity, all input images were cropped to a fixed size of 384×384 pixels "  This seems like a very ad-hoc number of pixels. How was this chosen?  How do authors make sure that no stroke regions are left out by this crop?


**Final Rating Justification:**

Authors have clarified all questions I had

**Justification Of The Preliminary Rating:**

The paper is very interesting and promising, but I have some concerns about the results.  If this was a application-specific issue, it should be easy to fix in the rebuttal. But it could also be that the method does not work well enough and then the paper is much less interesting.

**Paper Type:**

validation/application paper

**Questions To Address In The Rebuttal:**

Authors need to put the results in the context of what is clinically acceptable, since the reported DSC look very low: <0.35, while typically segmentation methods report results in excess of 0.6. This may be alright for this specific application, so I ask the authors to clarify this.

**Special Issue:**

no

---

> ### Author Response · Authors · 2021-03-18
> **Response to Reviewer 1**
>
> We would like to thank the reviewer for taking the time to review this submission and for providing a positive assessment of our work. Please find below point-by-point responses to each comment.
>
> ---
> **1. The reported DSC values seem very low (the highest is 0.33), while typical segmentation methods report results of more than 0.6.**
>
> We agree with the reviewer that the achieved Dice value (DSC=0.33) might seem low compared to those reported for other medical image segmentation tasks. Nevertheless, for the specific application of stroke lesion outcome prediction, a DSC of 0.33 is in the range of previously published results ([Pinto et al., 2018a](https://doi.org/10.3389/fneur.2018.01060); [Winzeck et al., 2018](https://doi.org/10.3389/fneur.2018.00679)), although not directly comparable due to different experimental setups and databases. A DSC of 0.33 also highlights that this is a challenging problem for which 4D CTP has predictive potential. For example, on a segmentation task, the input image already contains all relevant information to delineate the stroke lesion. However, for a lesion outcome prediction problem, the ground truth lesion is defined on images acquired at a later time point and many aspects can influence the dynamic and complex lesion growth.
>
> ---
> **2. The results show that using longer sequences improves predictions, however, there is no justification that even the best performing results are good enough.**
>
> We agree with the reviewer that judging whether results in terms of Dice values are good enough in a clinical context is hard given these preliminary results. In practice, this will depend on many aspects related to the actual case and the exact brain region being affected. Generally, a true clinical benefit will have to be shown within a prospective study. However, we believe that such a prospective study should only be conducted using the best-performing model. Thus, the current method is a step in the right direction. Furthermore, assessing and ranking stroke outcome prediction models via Dice scores is an established practice in our community (see for example the results of the [MICCAI ISLES 2016/2017 challenges](https://www.ncbi.nlm.nih.gov/pmc/articles/PMC6146088/); esp. Tab.7)
>
> ---
> **3.  “… all input images were cropped to a fixed size of 384x384 pixels.” How was this chosen?**
>
> To determine the optimal fixed size, we first generated a bounding box to identify the brain region in each patient. In doing so, we found that 384x384 is the smallest squared size in which all previously identified boundary boxes are able to fit in, ensuring that no stroke regions are left out from the analysis. Therefore, the brain region delineated by the bounding box was extracted and then zero-padded to fit this desired dimension.

---

> > ### Comment · AnonReviewer1 · 2021-03-18
> > **Response to rebuttal**
> >
> > Thanks for the clarifications

---

### Official Review · AnonReviewer2 · 2021-03-05

**Confidence:** 4
**Preliminary Rating:** 2

**Summary:**

The authors predict the final infarct area from a time series of contrast-enhanced CT perfusion scans. They propose a so-called Temporal Conv Net structure derived from a cited publication, and adapted to their use case. They then train and evaluate it against a gold standard segmentation of the stroke area. The metrics reported are DSC and volume error.

**Strengths:**

It is a valuable undertaking to address the prediction of infarct development with methods that don't require the determination of an AIF (often manual, subject to artefacts, and causing complex calculations afterwards).

The paper presents this need clearly and sets out to tackle the problem with a custom CNN architecture.

There is a sufficiently large dataset available for this research, including an annotated target.

The authors demonstrate familiarity with the clinical setting and can look at own prior work that helps them to get rid of several problems of data preprocessing.

The paper is written with a clarity that makes it sufficiently easy to follow the path (apart from the weaknesses addressed next).

**Weaknesses:**

* The paper lacks clarity in many parts
(1) What is the clinical workflow in terms of time? "Time is brain", but on which temporal scale? Seconds? Minutes? Would CT scans be watched interactively during contrast inflow, and the scanning be interrupted after some time if there is clearity? Or do they run through at all times, and are then analyzed?
(2) What in the context means "outcome prediction"? This is the overarching theme and title of the paper, but it is nowhere defined. Is it finally just a segmentation? From the final figure it appears there are follow-up scans (taken when?) on which the GT is annotated (by whom?)
(3) Unclear predictive goal (and perhaps unclear clinical motivation) makes it hard to assess what is written thereafter. In 3.1.1, the authors make a point about the causal convolutions required to ensure no data leaks from past to future. Is therefore the goal to predict the infarct shape at the end of the scan? I would not have assumed this, as we are looking at 60 sec scans, and I cannot imagine that a patient is rapidly pulled out of the CT and treated if within these 60 sec some predictive algorithm is able to foresee a malignant development of the infarct. Then, if in any case the full set of temporal images is available for future patients, there is no reason to account for the temporal nature of the data explicitly, and it could just as well be modelled as a 4D dataset.
(4) Unclear gold standard (I): Is it the final infarct area/volume? And if so: measured how, and when? Or is it the area/volume seen from the 60 second temporal contrast CT? And again: when? Looking at the comparison of the "gold standard" with model prediction (Fig. 2), there is a vast difference between the two.
(5) Unclear gold standard (II): Who and based on which criteria and with which tool annotated the gold standard? It apparently vastly differs from the segmentation.

* Why is it modelled using 2D networks? I don't see a convincing reason for this given by the authors. I hypothesize that there is large potential to improve the results by moving to 3D/4D architectures. In particular, a 2D (plus time) approach like it is presented cannot estimate the AIF (or a surrogate thereof) since the relevant information like e.g. an artery isn't a priorily spatially close to the infarct area. The authors defer the 3D approach to upcoming research, but I think it is a fundamental weakness to tackle a problem knowingly with methods that are not suited optimally. As the authors have suitable GPUs at their disposal, this is even more pressing.
* The work is not compared to any baseline (ConvLSTM, to just name the most obvious example, but even some optimized U-Net would be interesting). Instead, an intrinsic comparison is given to their proposed network, just pruned by removing the temporal information integration path. This will definitely impair the competitor substantially, since the architecture is specifically designed around the temporal component, so that I don't see any value in this comparison.
* Why is it important to compare inference times? I would suspect that even 2 minutes are much faster than any human can look at the temporal data, and also much faster than comparison workflows that mark the AIF and calculate the perfusion map will be. Again, it would tremendously help to read these details on the backdrop of a clear clinical workflow description.
* The authors say that they use a Dice loss because the lesion is often a small area only. I doubt that this is a good reasoning. Particularly for small targets, the Dice Coefficient is notorious for its malfunction (if no counter-measures are taken).
* With a batch size of 1, how was training stabilized (BatchNorm not functional)?

**Deanonymize Review:**

no

**Detailed Comments:**

I think I was quite detailed above ;-)
I didn't notice any typos, therefore apologies if I leave this section blank.


**Justification Of The Preliminary Rating:**

Without a comparison to a gold standard method it is hard to assess the added clinical predictive/diagnostic value, in particular as the quantitative metrics don't really look very good for themselves.
Judged for a methodological development, however, there are substantial lacks in the motivation of the approach (2D, temporal forward-only).
I hope that some of these concerns can be addressed in a rebuttal. I see the potential value of the undertaking.

**Paper Type:**

validation/application paper

**Questions To Address In The Rebuttal:**

I would rather read this publication again (and then potentially accept) after additional research answering the questions in the "weaknesses" section above has been conducted (3D approach, comparative implementations that are really competetive).
Most importantly, the exact definition of the modelling goal needs to be given, compare comments above. It needs to be clarified what exactly is predicted, and how this is more than a segmentation.

In terms of writing etc. there isn't much to ask for.

**Special Issue:**

no

---

> ### Author Response · Authors · 2021-03-18
> **Response to Reviewer 2  (Part 1)**
>
> We would like to thank the reviewer for taking the time to review the manuscript and for providing detailed comments and suggestions to improve the paper. Please find below point-by-point responses to each comment.
>
> ---
> **1. What is the clinical workflow in terms of time?**
>
> Initial triage begins with the early recognition of stroke patients. Once an acute stroke is suspected, patients typically undergo a non-contrast CT scan to exclude an intracranial hemorrhage. If no hemorrhagic signs are detected, CTP imaging is typically performed (for about 60 seconds) and then post-processed to produce perfusion parameter maps, which allow the evaluation of brain tissue status and are often used for clinical decision making. Based on clinical evaluation and interpretation of cerebral imaging, the patient’s eligibility is determined for appropriate stroke intervention and triage to tertiary health care centers. More details on acute stroke patient management can be found in [Waqas et al. (2019)](https://doi.org/10.1093/neuros/nyz067).
>
> It is worth noting that, for each hour in which stroke is untreated, the brain loses as many neurons as it does in nearly 4 years of natural aging. Therefore, the utilization of automatic image analytics software is encouraged, as it allows a quantitative assessment of the brain tissue status immediately after the CTP imaging is completed, more rapidly than human interpretation.
>
> ---
> **2. What in the context means outcome prediction? Is it finally just a lesion segmentation?**
>
> From the time of CTP imaging, the lesion typically grows until blood flow through the blocked artery is restored. This lesion growth is a highly dynamic and complex process that is affected by several parameters. Thus, stroke lesion outcome prediction consists of predicting follow-up changes in the anatomical extent of lesions over time, using the imaging acquired at baseline (i.e., CTP). To perform this prediction, our method assigns a probability of being part of the follow-up lesion outcome to each voxel of the image. A binary mask is then generated from these probability maps (lesion segmentation) to determine the volume and evaluate the outcome prediction although, in clinical practice, the actual “risk map” displaying the voxel-wise lesion probabilities is also of high interest.
>
> ---
> **3. Unclear predictive goal (and perhaps unclear clinical motivation) makes it hard to assess what is written thereafter. Is therefore the goal to predict the infarct shape at the end of the scan?**
>
> Our goal is to predict the lesion shape (i.e., at a 7-day follow-up) from baseline CTP images. As mentioned above, many factors influence the lesion growth with the actual treatment approach and its success as well as the time being some of the most important variables. In this study, predictions were performed and evaluated using retrospectively collected data with known outcomes. However, in a clinical scenario, a prediction would take place upon completion of the scan to estimate the tissue outcomes for different treatment options. An accurate stroke lesion outcome prediction method is of high interest, as it can potentially aid clinicians in the treatment decision-making process in a timely fashion.
>
> To help clarify the predictive goal, we have added the following sentence to the introduction: *“... predict the final infarct from perfusion maps. The basic idea of these methods is to train a model, using perfusion maps acquired at baseline prior to any treatment and their corresponding follow-up images acquired several days after treatment, to predict the final lesion location and volume in new patients.”*
>
> ---
> **4. Unclear gold standard. Is it the final infarct area/volume? Or is it the area/volume seen from the 60-second temporal contrast CT? Who, based on which criteria, and which tool annotated the gold standard?**
>
> The ground truth used in this work is the final infarct area segmented in follow-up imaging. We have updated the manuscript to address this concern  (section 4.1 - Patient data), and the new sentence describing the gold standard reads as follows: *“Follow-up scans (based on either non-contrast CT or MR) were acquired between 30 hours and seven days after stroke symptom onset. The follow-up lesions of each dataset were manually segmented semi-automatically by different experienced medical experts using AnToNIa (Forkert et al., 2014) and ITK-SNAP (Yushkevich et al., 2006) software tools.”*

---

> ### Author Response · Authors · 2021-03-18
> **Response to Reviewer 2 (Part 2)**
>
> **Part 2 - Please see Part 1 first.**
>
> ---
> **5. Why is it modelled using 2D networks?**
>
> We appreciate the reviewer’s suggestion and agree that results could potentially improve by moving to a 3D + time architecture since information within a 3D neighborhood may be leveraged at a deep level. However, we believe that given our current data with an inter-slice spacing of 5 mm, using the third spatial dimension does not add much useful information for solving this problem. Furthermore, using 3D networks comes with several disadvantages over 2D networks, such as higher computational costs and lower inference speeds. Also, to learn meaningful representations, an increased number of samples is required to train the larger number of parameters in 3D networks. Therefore, given the limited GPU memory and dataset available, we decided to restrict our analysis in this paper to 2D slices.
>
> Current state-of-the-art approaches ([Nielsen et al., 2018](https://doi.org/10.1161/STROKEAHA.117.019740); [Pinto et al., 2018a](https://doi.org/10.3389/fneur.2018.01060)) mainly use 2D networks to predict the stroke lesion outcome, further encouraging the use of 2D networks for the present study. Nevertheless, we recognize that additional details supporting the use of a 2D approach are needed, so we have added the following sentence to section 3 - Methods: *“... less computationally expensive compared to using 3D volumetric data (Singh et al., 2020). Current state-of-the-art approaches (Nielsen et al., 2018; Pinto et al. 2018a) use 2D networks to predict the stroke lesion outcome, further encouraging the use of 2D slices for the present study”.*
>
> ---
> **6. The work is not compared to any baseline (i.e., ConvLSTM or some optimized U-Net). An intrinsic comparison as done in the paper might not be valuable.**
>
> We thank the reviewer for suggesting the use of other architectures capable of handling temporal information such as ConvLSTM, and we will consider those approaches for follow-up papers. However, we are not aware of any stroke outcome prediction papers using 4D CTP image data in conjunction with LSTMs or RNNs that have been published yet. We, furthermore, respectfully disagree that the intrinsic comparison carried out as described in the paper may not be valuable. We consider our baseline encoder-decoder architecture (named ‘E-D’ in Tab. 1 & 2) to be equivalent to an (optimized) U-Net and, hence, believe that our results nicely illustrate the positive effect that using real temporal information has for the tissue outcome prediction problem in acute ischemic stroke patients.
>
> ---
> **7.  Why is it important to compare inference times?**
>
> As mentioned in section 1 - Introduction, *“the longer therapy is delayed, and blood flow is not restored, the more tissue may become irreversibly damaged”*. Therefore, faster inference times and thus earlier treatment decision is crucial for the efficient management of acute stroke patients.
>
> ---
> **8. Authors say that they use a Dice loss because the lesion is often a small area only. I doubt this is a good reasoning. Particularly for small targets, the Dice Coefficient is notorious for its malfunction.**
>
> The reviewer is correct regarding the high sensitivity of the Dice loss to small targets compared to large ones. However, the Dice loss is one of the most commonly used functions within the medical imaging community, especially for training neural networks for semantic image segmentation tasks ([Taghanaki et al., 2021](https://doi.org/10.1007/s10462-020-09854-1)). Initially introduced by [Milletari et al. (2016)](https://doi.org/10.1109/3DV.2016.79), a Dice loss is mainly implemented to tackle the class imbalance problem often found in medical image datasets, as it can easily learn from classes with lesser spatial representation in an image. Although we agree that it is not optimal, there is a lack of more appropriate loss functions for this specific problem.
>
> ---
> **9. With a batch size of 1, how was training stabilized (BatchNorm not functional)?**
>
> Before training the network, the input images were standardized by normalizing them to a zero-mean, unit-variance space across all selected time points. Apart from this, we implemented an L2 regularization and a Dropout of 0.5 to the end of the deepest convolutional layers to prevent overfitting. The step-based learning rate decay that was used also helped stabilize the initial training phase, which led to better training performances.

---

### Official Review · ~Matthan_W._A._Caan1 · 2021-03-08

**Confidence:** 5
**Preliminary Rating:** 3
**Recommendation:** Oral, Poster

**Summary:**

This paper presents a new temporal convolutional network based method to segment follow-up infarct lesions based on acute phase 4D CT perfusion data with the use of CT timeframes. Although this task has been optimized before as described in the related work section, this method is much simpler and aims to identify the optimal number of CT timeframes required for a deep learning model to segment the follow-up lesion. Performance is compared with a conventional method that uses perfusion maps extracted with proprietary software from the CT time series. The proposed method does outperform existing methods by means of the dice similarity coefficient.


**Strengths:**

The paper includes a thorough prior work section of previous techniques that used CTP data for estimating follow-up lesion. The motivation of this study is clearly described. Since the proposed method is simpler than existing methods it is interesting to know the segmentation performance.

**Weaknesses:**

The model size is largely increasing with the number of time steps. Can the improvement in DSC not be attributed to model size (and thus number of parameters) alone? Also, it is known that the tail of the time-intensity curve is of marginal added value, so that marginal improvement might be expected for 32T. These points should at least be mentioned in your discussion section and are contradicting your following statements:
Discussion/results: “Overall, these experiments demonstrate that increasing the time window size is beneficial, which is expected since including more time points provides the model with more information. Therefore, we can assume that the proposed architecture can make better use of the spatio-temporal information available in 4D CTP datasets to improve the stroke outcome predictions, especially when compared to using the simple perfusion maps for tissue outcome prediction.”
Conclusion: “The results demonstrate that using longer sequences improves the final infarct prediction, implying that the proposed model effectively learns from the spatio-temporal information provided and leads to better results compared to using simple perfusion maps.”
It is unclear how precisely the ground truth was obtained. How was the manual segmentation at follow-up created and, on what data (CT/MRI) was follow-up imaging it based, and how much time was there between baseline CTP and follow-up imaging? Furthermore, was ischemia treatment optimal so that no additional ischemic damage, and thus lesion growth, could occur between baseline CTP and follow-up imaging? Please explain.
Data from multiple centers were included with possibly varying scanning settings, e.g. the spatial/temporal resolution. Could this partly account for variability in reported results?
It remains unclear with what software and what settings the perfusion maps were computed.


**Deanonymize Review:**

yes

**Detailed Comments:**

Please list the training curve with training- and validation- set errors to confirm that training has converged at the chosen number of epochs. Do you see faster convergence for 8T and possible overtraining for 32T?

**Justification Of The Preliminary Rating:**

A nice network structure for temporal modelling is introduced. Currently, it cannot be judged whether the seen improvement is due to the larger network size or the broader temporal window as claimed by the authors.

**Paper Type:**

methodological development

**Special Issue:**

no

---

> ### Author Response · Authors · 2021-03-18
> **Response to Reviewer 4 (Part 1)**
>
> We would like to thank the reviewer for taking the time to assess our manuscript, as well as for providing positive comments and raising thoughtful points. Please find below point-by-point responses to each comment.
>
> ---
> **1. The model size is largely increasing with the number of time steps. Can the improvement in DSC not be attributed to the model size alone?**
>
> We thank the reviewer for raising such an interesting point. It is true that the number of trainable parameters considerably increases with the number of time points included in the sequence. This increase in model size is primarily due to the number of encoders that are generated to independently map the image of each time point to a latent space. That being said, one cannot exist without the other. However, we do believe that future optimizations can be done to the model (i.e., weight-sharing) to decrease the model size without affecting its performance.
>
> ---
> **2. It is unclear how precisely the ground truth was obtained.**
>
> We thank the reviewer for pointing this out. We have updated the manuscript to address this concern, and the new sentence describing the ground truth reads as follows: *“Follow-up scans (based on either non-contrast CT or MR) were acquired between 30 hours and seven days after stroke symptom onset. The follow-up lesions of each dataset were manually segmented semi-automatically by different experienced medical experts using AnToNIa (Forkert et al., 2014) and ITK-SNAP (Yushkevich et al., 2006) software tools.”*
>
> ---
> **3. Was the ischemia treatment optimal so that no additional ischemic damage, and thus lesion growth, could occur between baseline CTP and follow-up imaging?**
>
>  There will always be ischemic damage to some extent that occurs between baseline and follow-up imaging since the therapeutic intervention is provided after the baseline imaging has been performed. Lesion growth depends on several factors such as the type of treatment, the success of the treatment (i.e. recanalization and reperfusion success), and the time from baseline imaging to treatment. The proposed model aims to predict this lesion growth, which includes the additional ischemic damage that occurs between baseline and follow-up imaging. That being said, we ensured that no patients with ischemic damage that is unrelated to the acute perfusion deficit, such as patients with large prior stroke or hemorrhage, are included in this study.
>
> We have added the following sentence (section 1 - Introduction) to provide more details regarding this subject: *“The observed lesion growth is a result of several factors such as type of treatment, success of the treatment, and the time from baseline imaging to treatment.”*
>
> ---
> **4. Data from multiple centers were including with possibly varying scanning settings. Could this partly account for variability in the reported results?**
>
> We agree with the reviewer that this might contribute to the overall variability. However, one may argue that CTP is somewhat quantitative (linear dependency of the signal to the contrast agent density). Additionally, we used a validated processing pipeline, which further normalizes all CTP scans to a temporal resolution of 1 frame per second using a b-spline approximation of the time density curves. Nevertheless, different reconstruction kernels and contrast agent injection protocols might still affect the CTP datasets. We do believe that the multi-center nature of this database makes the developed model more generalizable and applicable in other health care centers, which we consider a benefit rather than a limitation. However, we do agree that the results would be most likely better if single-center datasets would have been used.
>
> ---
> **5. It remains unclear with what software and what settings the perfusion maps were computed.**
>
> The perfusion maps of CBV, CBF, MTT, and Tmax were computed based on deconvolved time-concentration curves using the software tool AnToNIa ([Forkert et al., 2014](https://doi.org/10.3414/me14-01-0007)). Briefly described, an arterial input function (AIF) was selected from the contralateral MCA or ICA using an atlas-based approach. This AIF was then used to compute voxel-wise residue functions using a block-circulant singular value decomposition of the concentration-time curve with the AIF employing a threshold of 0.15. Finally, the four perfusion parameters were determined for each voxel using the resulting residue function. This is a standard approach for perfusion analysis, whereas this perfusion analysis software has been used in various studies to compute perfusion maps from acute stroke CTP or PWI data.
>
> We have added the following sentence to the manuscript (section 4.3 - Implementation details) to clarify this aspect: *“... trained using perfusion maps (CBF, CBV, MTT, Tmax) and the CTP baseline average as input. These perfusion maps were obtained by deconvolving the time-concentration curve in each voxel with the AIF using AnToNIa.”*

---

> ### Author Response · Authors · 2021-03-18
> **Response to Reviewer 4 (Part 2)**
>
> **Part 2 - Please see Part 1 first.**
>
> ---
> **6. Please list the training curve with training- and validation- set errors to confirm that training has converged at the chosen number of epochs. Do you see faster convergence for 8T and possible overtraining for 32T?**
>
> We have computed the training curve for the 8T and 32T models for easier visual comparison: [training curves](https://uofc-my.sharepoint.com/:i:/g/personal/kimberlyalejandra_am_ucalgary_ca/EUlDfK9vlw9BhQbZtz8vdmwBXxsUXHw0vqrPg7Eye-vuQQ?e=PRqVWy).
>
> As observed in the image, the training- and validation-set errors for both models follow almost exactly the same trend, achieving convergence around epoch 120. Even though the loss for the 32T model starts at a higher point, it does achieve a lower loss throughout the rest of the training compared to the 8T model.

---

### Official Review · AnonReviewer3 · 2021-03-08

**Confidence:** 3
**Preliminary Rating:** 4
**Recommendation:** Oral, Poster
**Final Rating:** 4

**Summary:**

Authors present a method to predict the infarct core segmentation from 4D CTA images in a multi-centre data set. They demonstrate that their method outperforms a commonly used strategy based on perfusion maps. Additional experiments show that including more timepoints from the CTA yields better results, despite longer computation times.

**Strengths:**

To my understanding, this is an innovative method to use the full 4D CTA data to predict the infarct core segmentation on a follow-up image. Authors actually perform a prediction task: using the 4D CTA scan to predict the infarct core segmentation on a follow-up scan that is acquired at a later date. The use of a multi-centre data set is another strength. The experiments and evaluation is proper.

**Weaknesses:**

Based on the title and abstract, I was at first confused about the term "outcome prediction". I originally thought that the authors were going to determine patient outcome: e.g. future cognitive decline, dementia, or death. Only at the end of the introduction, it became clear that authors were aiming to determine the infarct core, based on the 4D CTA, but evaluated on a follow-up image of a later date. I think this should be clarified in the title and abstract.

To my opinion, the ratio of train / test subjects is a bit skewed. Do authors really need 133 patients to train on, leaving on 17 for test? The results would be more valuable if the test set could be increased (probably at the cost of reducing the training set, but I don't think that would impact the results much?).

After reading the paper, it is still not 100% clear to me if authors are segmenting just the infarct core or also the tissue-at-risk?

**Deanonymize Review:**

no

**Detailed Comments:**

Besides the points mentioned under weaknesses, I have some minor comments.

- can authors include the amount of time between the CTA and the follow-up images?
- can authors include a reference to the data set and/or mention which hospitals the data was acquired from?
- in the Discussion, it might be good to briefly compare the DSC / AVE values to comparable methods in literature. Some are mentioned in the Related Work section, but revisiting that in the Discussion would be helpful for the reader.

**Final Rating Justification:**

I'm happy with the changes the authors have made.

**Justification Of The Preliminary Rating:**

This is a nice and well-written manuscript. The rationale of the study is clear, method implementation is clearly described, and the experiments / evaluations performed are good. The results are good as well and there is a clear clinical relevance to this work.

**Paper Type:**

methodological development

**Special Issue:**

yes

---

> ### Author Response · Authors · 2021-03-18
> **Response to Reviewer 3**
>
> We would like to thank the reviewer for taking the time to review the manuscript and for providing suggestions that helped to further improve its quality. We sincerely appreciate the positive comments regarding the novelty and clinical relevance of the study. Please find below point-by-point responses to each comment.
>
> ---
> **1. The term “outcome prediction” should be clarified in the title and abstract.**
>
> We thank the reviewer for raising this point. We have added the following sentence to the abstract to clarify this aspect: *“... deep learning techniques to predict stroke lesion outcomes from perfusion maps. The basic idea is to train a CNN model using perfusion maps acquired at baseline prior to any treatment and their corresponding follow-up images acquired several days after treatment, to predict the final lesion location and volume in new patients.”*
>
> ---
> **2. The ratio of train/test subjects is a bit skewed. Results would be more valuable if the test set could be increased.**
>
> We agree with the reviewer that it would be valuable to increase the test set to better estimate the model performance. For this study, we ensured a similar distribution between the train and test sets using stratified sampling based on follow-up lesion volume. This means that, even though the test set is small, the results are representative of the entire population. Nonetheless, we will strongly consider increasing the size of the test set for follow-up papers.
>
> ---
> **3. Minor comments.**
>
> We have updated the manuscript to include the reviewer’s suggestions as follows:
>
> 1) Section 4.1 - Patient data: *“Follow-up scans (based on either non-contrast CT or MR) were acquired between 30 hours and seven days after stroke symptom onset. The follow-up lesions of each dataset were manually segmented semi-automatically by different experienced medical experts using AnToNIa (Forkert et al., 2014) and ITK-SNAP (Yushkevich et al., 2006) software tools.”*
>
> 2) Section 4.1 - Patient data: *“These scans were pooled together from the prospective cohort studies PRoveIT (Menon et al., 2015) and ERASER (Fiehler et al., 2019), as well as from data acquired at the University Medical Center Hamburg-Eppendorf, Germany, from June 2015 to May 2019.”*
>
> 3) Section 4.4 - Results and discussion: *“Although not directly comparable due to different experimental setups and databases, the Dice value achieved by the 32T model (DSC=0.33) is in the range of previously published results (Winzeck et al., 2018). Therefore, ...”*

---

> > ### Comment · AnonReviewer3 · 2021-03-18
> > **Response**
> >
> > Thanks for the clarifications made in the paper.

---

### Meta-Review · Area_Chair1 · 2021-03-28

**Recommendation:** Accept (Poster)

**Metareview:**

While the reviewers raised some concerns about the experimental results, I agree with the overall sentiment that this paper proposes a promising temporal convolutional network for segmentation in 4D CTP data. The authors may want to consider updating the title of the paper as the "outcome prediction" term seems to be a source of common confusion.

**Paper Type:**

both

---

### Decision · Program_Chairs · 2021-03-31

Accept